# Aortic Angiosarcoma Manifesting as Multiple Musculoskeletal Metastases: A Case Report

**DOI:** 10.3390/diagnostics14080805

**Published:** 2024-04-11

**Authors:** Won Jong Bahk, Sae Jung Na, In Yong Whang, Yongju Kim, Kyung Jin Seo

**Affiliations:** 1Department of Orthopedic Surgery, Uijeongbu St. Mary’s Hospital, College of Medicine, The Catholic University of Korea, Seoul 06591, Republic of Korea; wjbahk@catholic.ac.kr; 2Department of Radiology, Uijeongbu St. Mary’s Hospital, College of Medicine, The Catholic University of Korea, Seoul 06591, Republic of Korea; 3Department of Hospital Pathology, Uijeongbu St. Mary’s Hospital, College of Medicine, The Catholic University of Korea, Seoul 06591, Republic of Korea

**Keywords:** angiosarcoma, aorta, musculoskeletal metastasis, bone metastasis, positron emission tomography (PET)

## Abstract

Aortic angiosarcomas are rare. Due to its rarity and metastatic presentation, it is difficult to diagnose metastatic aortic angiosarcoma. We describe the clinicopathological and radiologic features of a metastatic aortic angiosarcoma presenting as musculoskeletal metastases. A 59-year-old male patient presented with left thigh pain. Plain radiographs revealed multifocal osteolytic lesions in the left femur shaft. Abdominopelvic computed tomography showed a lobulated osteolytic lesion in the left iliac bone. Magnetic resonance images revealed multifocal soft tissue lesions in the thigh musculature. A positron emission tomography/computed tomography (PET/CT) scan demonstrated multiple foci of increased uptake in the left femur bone, pelvis, left thigh, and calf musculature. Focal increased uptake in the lower abdominal aorta was newly detected. Pelvis biopsy showed tumor cell nests of epithelioid cells. The tumor cells showed vasoformative features. Immunohistochemically, the tumor cells showed positivity for vimentin, CD31, and ERG. The pathologic diagnosis of epithelioid angiosarcoma was established. The origin of the tumor was presumed to be the aorta. This case underscores the importance of PET scans in identifying primary lesions. In terms of the histopathologic diagnosis of biopsy samples with tumor cells exhibiting epithelioid neoplastic morphology, employing appropriate ancillary techniques such as immunocytochemistry with vascular markers may assist in accurately diagnosing metastatic angiosarcoma.

Angiosarcomas are rare sarcoma subtypes. They represent 2–4% of all sarcomas [1]. More than 50% of cases arise in the skin and seldom arise from the large vessels or the heart [2]. Aortic sarcomas are rare, and are most frequently seen in elastic arteries [3]. Among the aortic sarcomas, angiosarcomas are even rarer. In a case series of 26 cases of aortic sarcomas, only 8 cases were angiosarcoma. Among these 26 cases, 10 patients were found to have metastatic disease through imaging in various sites, such as the kidneys, skin or soft tissue, iliac bone, retroperitoneum, spine, brain, or adrenal gland. Less than 140 cases of aortic sarcoma have been reported with only 34 categorized as angiosarcoma [4]. Herein, we describe the clinicopathological and radiologic features of a metastatic aortic angiosarcoma presenting as multifocal musculoskeletal metastases of unknown primary.

A 59-year-old male patient presented with left thigh pain that had been increasing over the course of 2 weeks. Plain radiographs of both femurs revealed multifocal osteolytic lesions with occasional cortical thinning affecting the left femur shaft (Figure 1). Magnetic resonance (MR) images of the right thigh revealed multifocal soft tissue nodules in the anterior, posterior, and medial compartments of the thigh musculature (Figure 2a,b). These multiple ill-defined intramuscular nodules exhibit hypointense signal on both T1-weighted (Figure 2a) and T2-weighted images (Figure 2b) when compared to the adjacent musculature. The presence of ill-demarcated hyperintense signal around the nodules would suggest the infiltrative nature of tumor (Figure 2a,b, yellow arrows).

On MR, multiple bone lesions in the left femur shaft were hyperintense on both T1-weighted and T2-weighted images (Figure 3a,b).

On the abdominopelvic computed tomography (CT), an osteolytic lesion in the left iliac bone, 4 cm sized, is noted (Figure 4a). In addition, pelvic bone MR images revealed the detail of this left iliac bone osteolytic lesion (Figure 4b). This iliac lesion shows extraosseous extension.

In summary, we have detected multiple bone and soft tissue lesions involving the left femur and pelvic bone and the left thigh musculature. The radiologic differential diagnosis included metastatic carcinoma of unknown primary, multiple myeloma, and vascular malignancy such as epithelioid hemangioendothelioma or angiosarcoma. After discussion in a multidisciplinary tumor board meeting, an ^18^F-FDG PET/CT scan was performed and demonstrated multiple foci of increased uptake in the left femur bone, sacrum and left iliac wing of the pelvis, and left thigh and calf musculature (Figure 5a–c, yellow arrows). In addition to these lesions, focal uptake in the lower abdominal aorta was newly detected (Figure 5a, red arrow and Figure 6a–c, red arrows). We discussed this and decided to obtain a tissue sample from the left anterior superior iliac spine of the pelvis through incisional biopsy, to minimize tissue damage and complications.

Microscopic examination showed tumor cell nests in the bone marrow. The tumor cell nests were composed of tubules and solid nests of epithelioid cells (Figure 7a). Red blood cells were commonly observed amid the neoplastic cells. Areas displaying a vascular pattern consisted of irregular and sometimes widened vascular spaces filled with red blood cells. At higher magnification, the tumor cells exhibited large and epithelioid features, characterized by abundant eosinophilic cytoplasm and distinct cell borders. The nuclei showed marked pleomorphism and were frequently hyperchromatic. Mitoses were evident. Prominent, large eosinophilic nucleoli were frequent, often surrounded by a clear halo, giving them an appearance akin to inclusions. In some areas, the tumor cell nests were composed of mainly solid nests of plasmacytoid cells. Immunohistochemical stains showed that neoplastic cells were negative for CK, and positive for vimentin and vascular markers such as CD31 (Figure 7b) and ERG (Figure 7c). The Ki-67 index was more than 70% (Figure 7d). Based on these histopathologic and immunohistochemical stain findings, a pathologic diagnosis of metastatic epithelioid angiosarcoma was established. The origin of the tumor was presumed to be the aorta, based on the radiologic findings.

After the diagnosis was established, he underwent chemotherapy along with radiation therapy. Unfortunately, after four months, the patient’s dyspnea and fever worsened and he passed away.

Aortic sarcomas are rare, and angiosarcomas are even rarer [3]. In a case series of 26 cases of aortic sarcomas, only 8 cases were angiosarcoma. Among these 26 cases, 10 patients were found to have metastatic disease through imaging, and it was found in various metastatic sites. Less than 140 cases of aortic sarcoma have been documented, with only 34 identified as angiosarcoma [4].

From a radiologist’s point of view, it is very hard to differentiate vascular tumors of the musculoskeletal areas based on imaging findings, and clinical correlation can help narrow the differential diagnosis [5]. Angiosarcomas originating in bone are exceedingly rare, constituting less than 1% of primary malignant bone tumors [6]. Based on the findings from one European retrospective study, primary angiosarcomas of bone constitute 56%, while metastatic angiosarcomas of bone represent 44% of the total 80 patient diagnosed with angiosarcomas of bone [7]. The predominant sites for bone angiosarcomas are typically the long bones (60%), notably the tibia (23%), femur (18%), humerus (13%), and pelvis (7%) [5,6]. Radiographic and CT imaging of malignant vascular bone tumors exhibit diverse characteristics, often appearing lytic with poorly defined margins [8]. In a retrospective analysis involving 63 patients with malignant vascular bone lesions, multifocal lesions were observed in 40% of cases. However, the presence of multifocal lesions devoid of periosteal reaction could also suggest lytic bony metastases or multiple myeloma, especially in individuals aged over 40, warranting histopathologic confirmation in the diagnostic process [6,9]. It has been suggested that in individuals over 40 years old presenting with a well-defined osteolytic lesion exceeding 4 cm in the femur, featuring geographic cortical destruction and lacking evidence of periosteal reaction, suspicion should be directed towards a malignant vascular bone lesion [6,9].

In our case, the radiological differential diagnoses of multifocal lesions involving musculoskeletal system include metastases and multiple myeloma. Interestingly, a subsequent ^18^F-FDG PET/CT scan could identify the primary tumor origin. The primary tumor was not identified on the abdominopelvic CT. We misinterpreted the mural tumor of the aorta as mural thrombus with calcified atherosclerosis on the CT.

This case highlights two noteworthy points. Firstly, it underscores the significance of ^18^F-FDG PET/CT scans in detecting primary lesions, especially in cases where multiple metastatic lesions in the musculoskeletal system are suspected. Secondly, when the pathological examination reveals tumor cells of epithelioid neoplastic morphology, it can pose a diagnostic challenge for pathologists, especially in biopsy samples. Understanding the clinical setting and being aware of the morphologic clues like the vasoformative features of angiosarcoma, along with employing ancillary techniques such as immunocytochemistry with vascular markers (CD31, CD34, or ERG), may aid in establishing a pathologic diagnosis of metastatic angiosarcoma in bone biopsies.

## Figures and Tables

**Figure 1 diagnostics-14-00805-f001:**
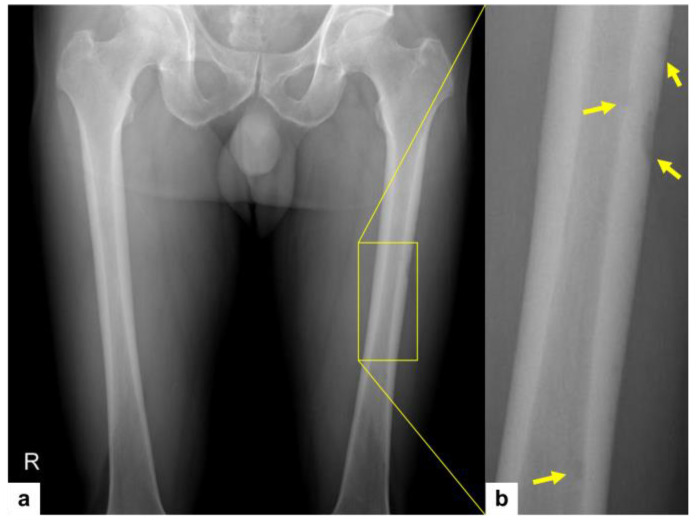
Plain radiograph of both femurs revealing multifocal lesions along the left femur shaft (**a**, the yellow box indicating the region of interest, ROI). Note that the osteolytic nature of these lesions, along with occasional cortical thinning, becomes evident on zooming into the ROI (**b**, yellow arrows).

**Figure 2 diagnostics-14-00805-f002:**
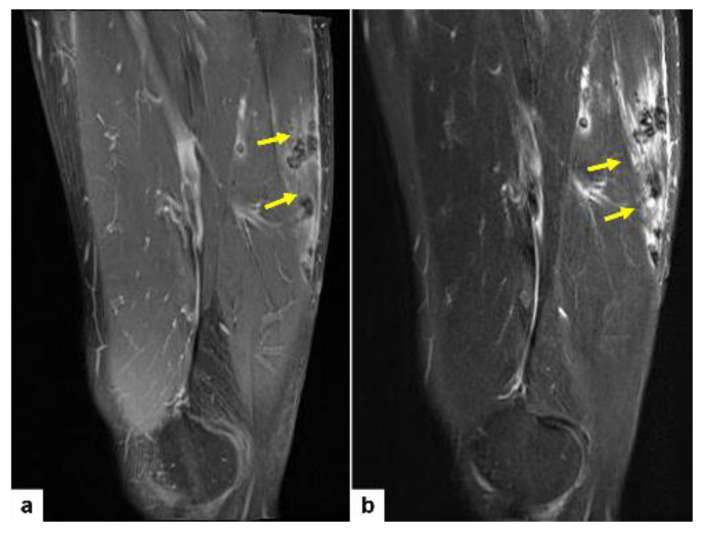
Coronal T1-weighted (**a**) and T2-weighted (**b**) MR images showing multiple ill-defined intramuscular nodularities in the right thigh. These nodularities show low signal intensity on both T1- and T2-weighted images, compared to adjacent musculature. Note peritumoral high signals suggesting infiltrative nature of tumor (**a**,**b**, yellow arrows).

**Figure 3 diagnostics-14-00805-f003:**
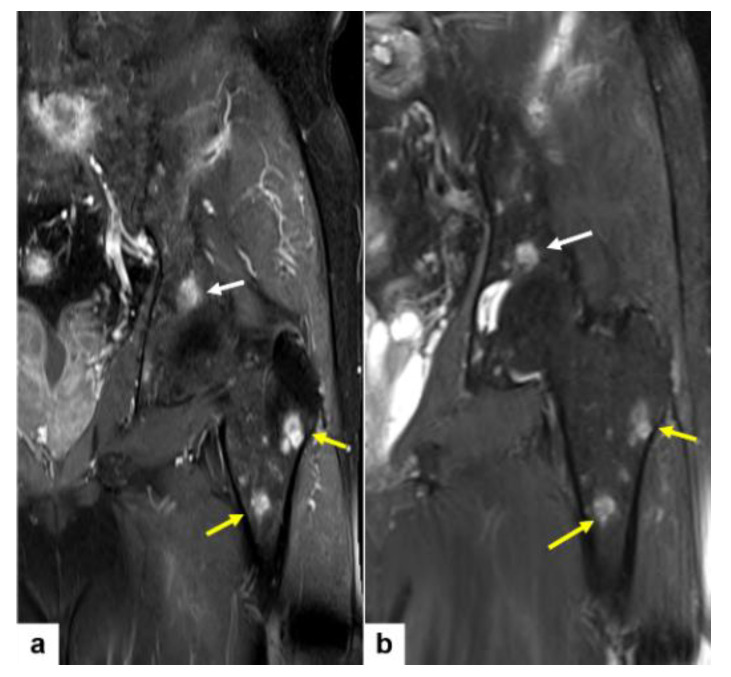
Fat-suppressed contrast-enhanced T1-weighted (**a**) and fat-suppressed T2-weighted (**b**) MR images of the pelvic bone reveal multiple hyperintense bone lesions in the left femur shaft (**a**,**b**, yellow arrows) and the left iliac bone (**a**,**b**, white arrows).

**Figure 4 diagnostics-14-00805-f004:**
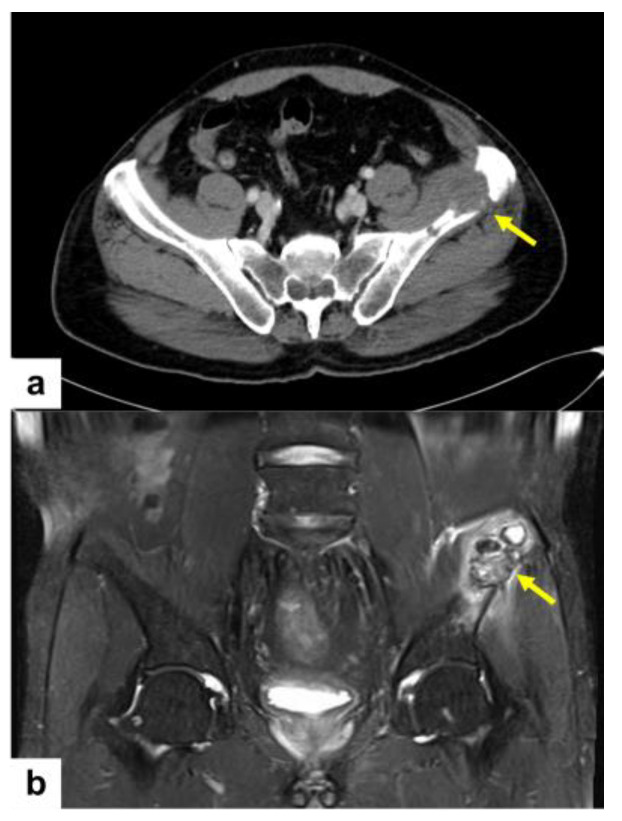
Abdominopelvic CT showing a 4 cm sized osteolytic lesion in the left iliac bone (**a**, yellow arrow). Fat-suppressed T2-weighted MR image of the pelvis reveals extraosseous extension (**b**, yellow arrow).

**Figure 5 diagnostics-14-00805-f005:**
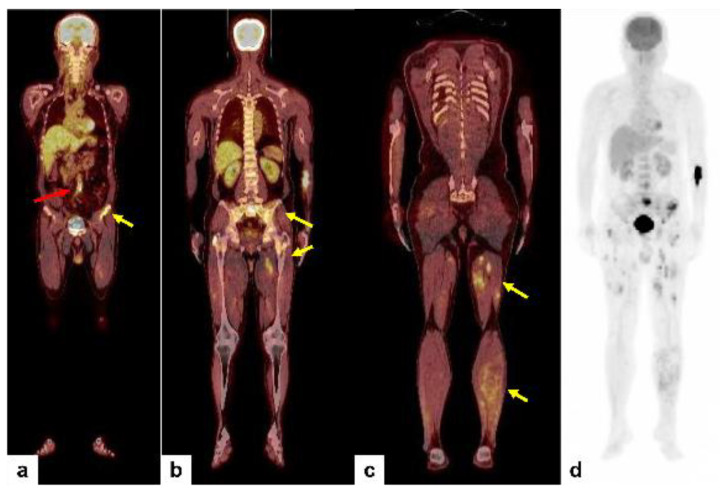
^18^F-FDG PET/CT scan was performed and demonstrated multiple foci of increased uptake in the left femur bone, sacrum and left iliac wing of the pelvis, and left thigh and calf musculature (**a**–**c**, yellow arrows; **d**, maximal intensity projection). Please note focal uptake in the lower abdominal aorta (max SUV 5.5) (**a**, red arrow).

**Figure 6 diagnostics-14-00805-f006:**
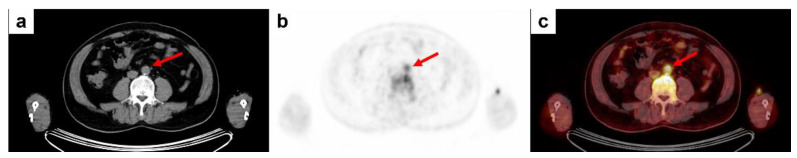
Axial CT (**a**), ^18^F-FDG PET (**b**), and ^18^F-FDG PET/CT scan (**c**) showed focal uptake in the lower abdominal aorta indicating the primary tumor (**a**–**c**, red arrow).

**Figure 7 diagnostics-14-00805-f007:**
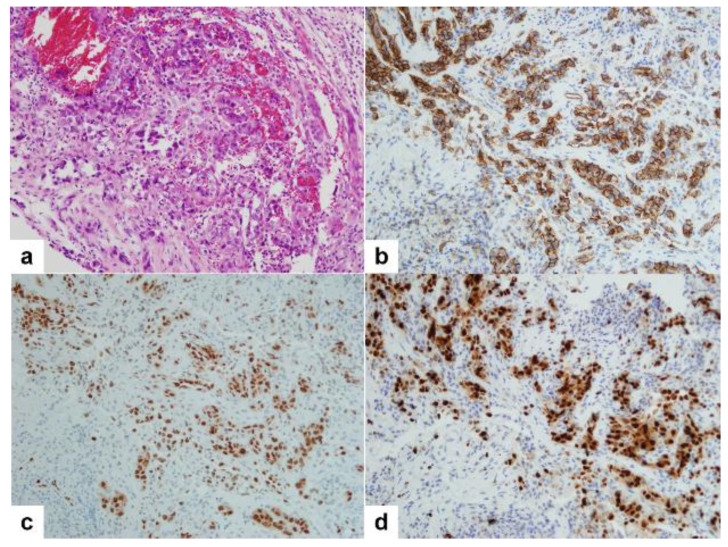
Microphotograph of the pelvic bone biopsy sample showing tumor cell nests composed of tubules and solid nests of epithelioid cells (**a**, H&E). Immunohistochemical stains showing positivity for vascular markers such as CD31 (**b**) and ERG (**c**). Tumor cells showing high Ki-67 index (70%) (**d**). (**a**–**d**, 200×).

## Data Availability

The data presented in this study are available upon request from the corresponding author.

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
