# Peer review of "Aortic Angiosarcoma Manifesting as Multiple Musculoskeletal Metastases: A Case Report"

_diagnostics, 2024, doi:10.3390/diagnostics14080805_

Round 1
Reviewer 1 Report
Comments and Suggestions for Authors
The authors reported an interesting case with a diagnosis of aortic angiosarcoma with multiple musculoskeletal metastases. The authors underscore the importance of FDG PET in locating the primary tumor in such cases. This case has some merit. My comments and suggestions are listed as follows,
1. The word count of the abstract exceeds the limit. The authors may reorganize the abstract to be more concise.
2. In this report, FDG PET is important in identifying the primary tumor. I would recommend adding an MIP (maximum intensity projection) to Figure 5. In addition, Fig. 5a may not be clear enough to highlight the primary tumor. I would recommend more detailed images to show the primary tumor. For example, axial PET, CT, and PET/CT with arrows.
3. The patient also underwent abdominopelvic CT (Figure 4a). Was the primary tumor identifiable on the abdominopelvic CT? If not, was the primary tumor out of the FOV (field-of-view) of CT? Or did the primary tumor not show an apparent structural abnormality on the CT? The readers of this article may be curious about why FDG PET identified the lesion, while the CT did not.
Author Response
Dear Reviewer,
Thank you for your meticulous review of our manuscript. We truly appreciate your valuable comments. The manuscript has been revised and appropriate changes have been made based on your insightful and helpful comments. The following text represents point-by-point responses to the comments. Thank you again for your consideration and excellent feedback.
Point-by-Point Responses:
Comments and Suggestions for Authors
The authors reported an interesting case with a diagnosis of aortic angiosarcoma with multiple musculoskeletal metastases. The authors underscore the importance of FDG PET in locating the primary tumor in such cases. This case has some merit. My comments and suggestions are listed as follows,
- The word count of the abstract exceeds the limit. The authors may reorganize the abstract to be more concise.
Response: We have reduced and reorganized the abstract.
- In this report, FDG PET is important in identifying the primary tumor. I would recommend adding an MIP (maximum intensity projection) to Figure 5. In addition, Fig. 5a may not be clear enough to highlight the primary tumor. I would recommend more detailed images to show the primary tumor. For example, axial PET, CT, and PET/CT with arrows.
Response: We have added an MIP (maximum intensity projection) to Figure 5. We have added axial PET, CT, and PET/CT with arrows.
- The patient also underwent abdominopelvic CT (Figure 4a). Was the primary tumor identifiable on the abdominopelvic CT? If not, was the primary tumor out of the FOV (field-of-view) of CT? Or did the primary tumor not show an apparent structural abnormality on the CT? The readers of this article may be curious about why FDG PET identified the lesion, while the CT did not.
Response: The primary tumor was not identified on the abdominopelvic CT. We have misinterpreted the mural tumor of the aorta as mural thrombus with calcified atherosclerosis on the CT.
Reviewer 2 Report
Comments and Suggestions for Authors
diagnostics-2936659 Unexpected primary tumor origin for multiple musculoskeletal metastases.
This is a case report of a rare malignant tumor, angiosarcoma originated in the abdominal aorta. The first patient’s complaint was left thigh pain, and FDG-PET imaging revealed aortic tumor along with multiple metastasis. Pathological diagnosis was confirmed by biopsy of the pelvic bone.
This is an important case report, and contains useful implications in clinical practice.
I have some concerns.
1. The title is not appropriate. The title should include “case report” and “angiosarcoma”. Although the title should reflect the content of this case report, the content of the text cannot be predicted from the current title. The title should not hint at the content, like a drama trailer.
2. There are many errors in the expression of the imaging.
1) PET examination demonstrated multiple hypermetabolic lesions in the bones and aorta. This might be FDG-PET??? In Figs. 5, the expression “hypermetabolic” is not appropriate. We usually use the term “increased uptake” or “focal uptake”.
2) There are many errors in the expression of MRI findings, which is unacceptable as is. For example, I have never seen the expression “signal loss” on conventional MRI. The expression “marked peritumoral edema indicative of vascular lesions” is also not acceptable. What finding was considered edema? Why did the authors think of finding “edema”? Why did the authors suspect “vascular lesion”? What is a “vascular lesion”? Did the authors mean hemangioma and AVM?
3) In the main text, “magnetic resonance imaging” should be expressed as “MRI”.
4) I cannot identify “lobulated” osteolytic lesions on CT (Figs.4).
5) In Figs. 5, the expression “hypermetabolic” is not appropriate. We usually use the term “increased uptake” or “focal uptake”.
The most important point in this case report is that image diagnosis greatly contributes to the diagnosis of the disease. Nevertheless, the authors did not include radiologists and therefore the description of the imaging diagnosis is extremely inadequate. Explanations about diagnostic imaging need to be completely rewritten.
Comments on the Quality of English Language
None
Author Response
Dear Reviewer,
Thank you for your meticulous review of our manuscript. We truly appreciate your valuable comments. The manuscript has been revised and appropriate changes have been made based on your insightful and helpful comments. The following text represents point-by-point responses to the comments. Thank you again for your consideration and excellent feedback. Your judicious comments have helped shape our manuscript into a better, more coherent version.
Point-by-Point Responses:
Comments and Suggestions for Authors
diagnostics-2936659 Unexpected primary tumor origin for multiple musculoskeletal metastases.
This is a case report of a rare malignant tumor, angiosarcoma originated in the abdominal aorta. The first patient’s complaint was left thigh pain, and FDG-PET imaging revealed aortic tumor along with multiple metastasis. Pathological diagnosis was confirmed by biopsy of the pelvic bone.
This is an important case report, and contains useful implications in clinical practice.
I have some concerns.
- The title is not appropriate. The title should include “case report” and “angiosarcoma”. Although the title should reflect the content of this case report, the content of the text cannot be predicted from the current title. The title should not hint at the content, like a drama trailer.
Response: We agree with your comment. We modified the title.
- There are many errors in the expression of the imaging.
1) PET examination demonstrated multiple hypermetabolic lesions in the bones and aorta. This might be FDG-PET??? In Figs. 5, the expression “hypermetabolic” is not appropriate. We usually use the term “increased uptake” or “focal uptake”.
Response: We changed the expression “hypermetabolic” as “increased uptake” or “focal uptake”.
2) There are many errors in the expression of MRI findings, which is unacceptable as is. For example, I have never seen the expression “signal loss” on conventional MRI. The expression “marked peritumoral edema indicative of vascular lesions” is also not acceptable. What finding was considered edema? Why did the authors think of finding “edema”? Why did the authors suspect “vascular lesion”? What is a “vascular lesion”? Did the authors mean hemangioma and AVM?
Response: We agree with your comment. We recruited colleague radiologists in revising process. With their participation, we corrected some awkward expressions such as “signal loss” and “marked peritumoral edema indicative of vascular lesions” and reinforced the description and discussion of radiologic findings.
3) In the main text, “magnetic resonance imaging” should be expressed as “MRI”.
Response: We changed as you mentioned.
4) I cannot identify “lobulated” osteolytic lesions on CT (Figs.4).
Response: We have changed CT image and have deleted the description of “lobulated” as you pointed out.
5) In Figs. 5, the expression “hypermetabolic” is not appropriate. We usually use the term “increased uptake” or “focal uptake”.
Response: We changed as you mentioned.
The most important point in this case report is that image diagnosis greatly contributes to the diagnosis of the disease. Nevertheless, the authors did not include radiologists and therefore the description of the imaging diagnosis is extremely inadequate. Explanations about diagnostic imaging need to be completely rewritten.
Response: We agree with your comment. We recruited colleague radiologists in revising process. With their participation, we revised and reinforced the description and discussion of radiologic findings.
Round 2
Reviewer 1 Report
Comments and Suggestions for Authors
Thank you for submitting the revised manuscript.
The authors have made substantial revisions to their manuscript and my concerns and suggestions have been addressed.
Author Response
Dear Reviewer,
Thank you for your meticulous review of our manuscript. We truly appreciate your valuable comments. We believe that the manuscript is substantially improved after making the suggested edits.
Reviewer 2 Report
Comments and Suggestions for Authors
Some minor corrections are needed.
Figure 2 caption : Eliminate "scan".
Figure 3 caption: This should be "fat-suppressed contrast-enhanced T1-weighted MR images", or "fat-suppressed T2-weighted images" (non-enhanced).
Figure 4: "computed tomography" should be changed to "CT".
In Discussion, "PET CT" should be "FDG-PET/CT".
Author Response
Comments and Suggestions for Authors
Some minor corrections are needed.
Response: Thank you for your meticulous review of our manuscript. We truly appreciate your valuable comments. We have revised as you mentioned.
Figure 2 caption : Eliminate "scan".
Response: We have eliminated the expression “scan”.
Figure 3 caption: This should be "fat-suppressed contrast-enhanced T1-weighted MR images", or "fat-suppressed T2-weighted images" (non-enhanced).
Response: We have changed the caption to “Fat-suppressed contrast-enhanced T1-weighted (a) and fat-suppressed T2-weighted (b) MR images”.
Figure 4: "computed tomography" should be changed to "CT".
Response: We have changed "computed tomography" to "CT".
In Discussion, "PET CT" should be "FDG-PET/CT".
Response: We have changed "PET CT" to "FDG-PET/CT".